# How much can language models memorize?

John X. Morris [2] [*]   Chawin Sitawarin [3] [*]   Chuan Guo [4] [*]   Narine Kokhlikyan [1]   G. Edward Suh [5]
Alexander M. Rush [2]   Kamalika Chaudhuri [3] [*]   Saeed Mahloujifar [1]

## Abstract

Due to the inherent structure of language, prior studies of language model memorization have struggled to disentangle memorization from generalization. We formally separate memorization into two components: *unintended memorization*, the information a model contains about a specific dataset, and *generalization*, the information a model contains about the true data-generation process. When we completely eliminate generalization, we can compute the total memorization, which provides an estimate of model capacity: our measurements estimate that GPT-style models have a capacity of approximately 3.6 bits per parameter. We train language models on datasets of increasing size and observe that models memorize until their capacity fills, at which point unintended memorization decreases as models begin to generalize. We train hundreds of transformer language models ranging from 500K to 1.5B parameters and produce a series of scaling laws relating model capacity and data size to membership inference.

## 1. Introduction

For the past several years, modern language models have been trained on increasingly large amounts of data, while parameter counts stay stagnant in the billions. For example, one recent state-of-the-art model (Dubey et al., 2024) has 8 billion parameters (around 32 GB on disk) but is trained on 15 trillion tokens (around 7 TB on disk).

A long line of work (Carlini et al., 2019; Mireshghallah et al., 2022; Nasr et al., 2023; Zhang et al., 2023; Carlini et al., 2023b; Schwarzschild et al., 2024) questions whether such pretrained language models memorize their training

data in a meaningful way. Most research approaches this problem either through the lens of extraction, aiming to recover full training data points from model weights, or membership inference, classifying whether a training point was present in the training data of a given model.

Studies of language model extraction argue that a data point is memorized if we can induce the model to generate it (Carlini et al., 2023b; Nasr et al., 2023; Schwarzschild et al., 2024). We argue that such generation does not necessarily serve as a proof of memorization. Language models can be coerced to output almost any string (Geiping et al., 2024); hence the fact that a model outputs something is not necessarily a sign of memorization.

We propose a novel definition of memorization that quantifies the extent to which a model retains information about a specific datapoint. Our approach leverages the concept of compression rate in bits, where a model is considered to have memorized an input if it can be compressed to a significantly shorter encoding in the presence of the model. This framework draws inspiration from Kolmogorov information (Kolmogorov, 1963) and Shannon information (Shannon, 1948), but can be easily measured in practice using model likelihoods. We tackle the fundamental challenge of distinguishing between memorization and generalization (Prashanth et al., 2024) by decomposing memorization into two distinct components: *unintended memorization*, which captures the information a model stores about a particular dataset, and *generalization*, which represents the knowledge a model has acquired about the underlying data-generating process.

To understand our new quantities, we measure unintended memorization and generalization by training language models of varying capacity on datasets of different sizes. We first eliminate the question of generalization entirely by training on a dataset of random uniformly-sampled bitstrings. In this setting, we can exactly measure the amount of information contained about the data inside the model. This gives us a principled way to measure language model *capacity* when trained on uniform datasets of exact known information content. We find that GPT-style transformers can store between 3.5 and 4 bits of information in each model parameter, depending on model architecture and precision.

---

[*]Work done while at Meta.   [1]FAIR at Meta [2]Cornell University [3]Google DeepMind [4]OpenAI [5]NVIDIA. Correspondence to: Saeed Mahloujifar <saeedm@meta.com>.

*Proceedings of the $43^{rd}$ International Conference on Machine Learning*, Seoul, South Korea. PMLR 306, 2026. Copyright 2026 by the author(s).

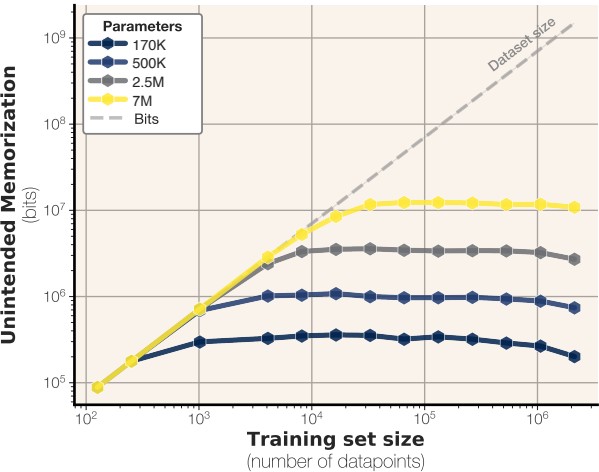

*Figure 1.* **Unintended memorization of uniform random data** (Section 3). Memorization plateaus at the capacity of different-sized models from the GPT-family.

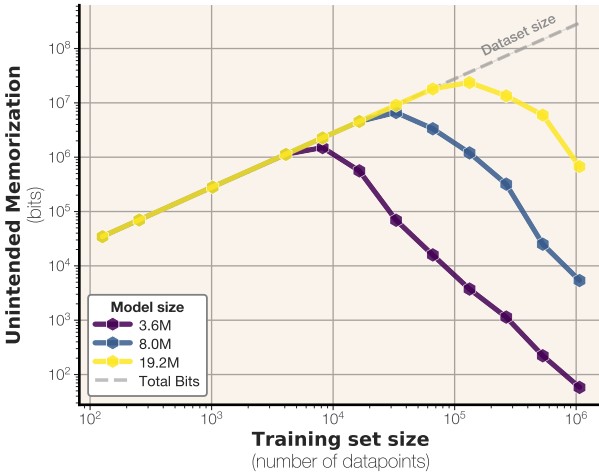

*Figure 2.* **Unintended memorization of text** across model and dataset sizes (Section 4). All quantities are calculated with respect to a large oracle model (1B params).

We then repeat our experiments with real text, where generalization is possible and even beneficial for learning. On real text, language models memorize up to a certain capacity, at which point they substitute unintended memorization for generalization, and begin to learn general, reusable patterns as opposed to sample-level specifics. Our framework shows that double descent begins to occur at this point, when the data size exceeds the model capacity in bits.

Finally, we use our results to predict a scaling law for membership inference performance based on model capacity and dataset size. We show that membership inference follows a clean relationship based on model capacity and dataset size: bigger models can memorize more samples, and making datasets bigger makes membership inference harder. Our scaling laws extrapolate to larger models, and predict most modern language models are trained on too much data to do reliable membership inference on the average data point.

## 2. Memorization, intended and unintended

When a model $\theta = L(x)$ is trained using a training algorithm $L$ and a dataset $x \sim X$, some information is transferred from the sample $x$ to the model $\theta$. A key question in the memorization literature is determining how much of this stored information is intended versus unintended. In this work, we aim to provide a rigorous definition of memorization that satisfies certain properties:

1. *Separation from generalization.* Our notion of unintended memorization must be distinct from intended memorization, which we refer to as generalization. For example, consider a language model trained on the sample: *Q: What is $2^{100}$? A: 1267650600228229401496703205376.* When assess-

ing how much of this training sample is memorized, we must account for the fact that performing simple math operations is expected from a language model.

2. *Sample-level memorization.* We need to define memorization for realizations of random variables, not the random variables themselves. Specifically, we want to determine how much unintended memorization of a sample $x$ occurs in a model $\theta$.

3. *Independence from training algorithm.* Our definition should be independent of the training algorithm $L$ and only a function of the final model $\theta$ and the sample $x$. This is crucial for language models, where we often only have access to the final model and target sample.

Previous works have attempted to define memorization for machine learning models. We aim to provide precise definitions of memorization that meet our criteria, and offer ways to measure it. See Appendix A.3 for a broader discussion on definitions of memorization.

**Roadmap for this section.** We first define memorization in terms of Shannon information (2.1): the definition is conceptually clean and operates over distributions, but as we will see it is not directly operationalizable from a single trained model and a single dataset. We then operationalize it in two steps. First (2.2) we switch to a Kolmogorov-complexity formulation that operates on individual strings; the two formulations agree in expectation (Proposition 4). Second (2.3) we describe how each Kolmogorov term is approximated in practice from model likelihoods. This is the end-to-end procedure that produces every $\text{MEM}_U$ number in this paper.

## 2.1. A statistical view of memorization

*Notation.* In this section, we use capital letters (e.g. $X$, $\Theta$) to refer to random variables and lowercase letters to refer to instances of a random variable (e.g. $x \sim X$ and $\theta \sim \Theta$).

Information theory has developed well understood notions of information for random variables. For a random variable $X$, we often use $H(X)$, the entropy of $X$, to define the amount of information present in $X$. Moreover, for two distinct random variables $X, Y$, we can define $H(X \mid Y)$ to be the uncertainty left in $X$ after fixing $Y$. Having defined this quantity, we can now measure *mutual information* between $X$ and $Y$ by subtracting the leftover information from the total information: $I(X;Y) = H(X) - H(X \mid Y)$.

Now assume we have a machine learning pipeline. We have a prior $\Theta$ on the underlying model that captures our dataset distribution $X$. And we have a learning algorithm $L$ that maps samples from $X$ to a trained model $\hat{\Theta}$. To understand how much information about $X$ is stored in $\hat{\Theta}$, we can use the notion of mutual information:

$$\text{mem}(X, \hat{\Theta}) = I(X; \hat{\Theta}) = H(X) - H(X \mid \hat{\Theta}).$$

Note that this captures all the information about $X$ that is stored in $\hat{\Theta}$. As we discussed, we need our notion of memorization to account for generalization as well.

**Example.** Recall the $2^{100}$ sample from the introduction, and contrast it with "John Smith scored 147 points in the 2019 regional bowling championship." A capable language model should be able to compute $2^{100}$ on its own, so a reference model $\Theta$ that captures the data distribution already explains the first sample, and our trained model $\hat{\Theta}$ provides little additional reduction in uncertainty about it. For the second, the reference can predict the syntactic frame but not the specific name, score, and year; $\hat{\Theta}$ reduces our uncertainty further. We want our definition to attribute the first to generalization and the second to memorization-of-the-sample. Formally, when measuring unintended memorization, we are only interested in the information that is present in $X \mid \Theta$, which is the uncertainty left in $X$ after fixing $\Theta$. Hence, we can define **unintended memorization** as

$$\text{mem}_U(X, \hat{\Theta}, \Theta) = I(X; \hat{\Theta} \mid \Theta)$$
$$= H(X \mid \Theta) - H(X \mid (\Theta, \hat{\Theta})).$$

and then the stored **generalization** (or intended memorization) must be

$$\text{mem}_I(\hat{\Theta}, X, \Theta) = \text{mem}(X, \Theta) - \text{mem}_U(X, \hat{\Theta}, \Theta)$$
$$= I(X; \hat{\Theta}) - I(X; \hat{\Theta} \mid \Theta)$$

Now that we have defined our notions of intended and unintended memorization we turn our attention to practically

measuring them. Let us first state a proposition that enables measurement of unintended memorization:

*Proposition* 1 (Super-additivity of Unintended Memorization). Assume $X = (X_1, \ldots, X_n)$ is a dataset of $n$ i.i.d. samples. We have

$$\sum_{i \in [n]} \text{mem}_U(X_i, \hat{\Theta}, \Theta) \leq \text{mem}_U(X, \hat{\Theta}, \Theta) \leq H(\hat{\Theta}).$$

This proposition shows that to measure a lower bound on the unintended memorization at the dataset level, we can sum per-sample memorization. The entropy of the information content of the trained model itself serves as an upper bound on the unintended memorization. This implies that unintended memorization should scale with the dataset size but cannot exceed the total capacity of the model.

## 2.2. Measuring unintended memorization with Kolmogorov Complexity

Our definitions of memorization and generalization so far are defined using an "entropy-based" notion of information. This means our definitions can only be used for random variables. This brings challenges in measuring memorization. All our variables in the definition of memorization are singletons. We have a single underlying model $\theta$, a single dataset $x = (x_1, \ldots, x_n)$ and a single trained model $\hat{\theta}$[1]. It is impossible to measure the entropy (let alone conditional entropy) of the underlying variables using a single sample.

To this end, we switch to another notion of information based on compression, then later we show how this notion closely approximates the notion of memorization defined above. Kolmogorov complexity defines the information content of a string $x$, denoted as $H^K(x)$, to be the length of shortest representation of $x$ in a given computational model. Similarly, we can define the leftover information $x \mid \theta$ to be the shortest representation of $x$ when we have $\theta$ available as a reference. The information content of $x \mid \theta$, denoted by $H^K(x \mid \theta)$, is the length of such description. We can then define mutual information in a similar fashion:

*Definition* 2 (Kolmogorov complexity). Let $f$ be an arbitrary computational model that takes a set of inputs and returns an output (e.g. universal Turing machine). The shortest description of $x$ with respect to computational model $f$ is defined as $H^K(x) = \min_{f(p)=x} |p|$. The Kolmogorov complexity of $x$ relative to another string $\theta$ is defined as $H^K(x \mid \theta) = \min_{f(p,\theta)=x} |p|$. We define the Kolmogorov mutual information between $x$ and $\theta$ by $I^K(x, \theta) = H^K(x) - H^K(x \mid \theta)$. We assume inputs are bitstrings and $|p|$ is the bit length of the input.

*Definition* 3 (Kolmogorov memorization). Let $\theta$ be a reference model that approximates the true distribution of data,

---

[1]Note the switch to lowercase variables because we are now working with instances, not random variables.

and $\hat{\theta}$ be a model trained on a dataset $x = (x_1, \ldots, x_n)$. For each $x_i$ we define the memorization of $x_i$ in $\hat{\theta}$ as $\mathrm{mem}^K(\hat{\theta}, x) = I^K(\hat{\theta}, x)$.

We also define intended and unintended variants of memorization:

$$\mathrm{mem}_U^K(x, \theta, \hat{\theta}) = H^K(x \mid \theta) - H^K(x \mid (\theta, \hat{\theta})),$$
$$\mathrm{mem}_I^K(x, \theta, \hat{\theta}) = \mathrm{mem}^K(x, \hat{\theta}) - \mathrm{mem}_U^K(x, \theta, \hat{\theta}).$$

There are known connections between Kolmogorov complexity and Shannon Entropy (Grunwald & Vitanyi, 2004). These results point at the conceptual connection between the two notions and imply that $\mathrm{E}_{x \sim X}[H^K(x)] \approx H(X)$. Interestingly, this implies that our notion of Kolmogorov memorization closely approximates Shannon memorization.

*Proposition* 4. Let $X = (X_1, \ldots, X_n)$ be an i.i.d. dataset distribution parametrized by ground-truth model $\theta$. Let $L$ be a training algorithm mapping $X$ to $\hat{\Theta}$. Assume $H(\hat{\Theta}) = \ell$ and $H(X_i) = \ell'^2$. Then we have $\left| \mathrm{E}_{\substack{x \sim X \\ \hat{\theta} \sim L(x)}} \left[ \mathrm{mem}_U^K(x_i, \hat{\theta}, \theta)] \right] - \mathrm{mem}_U(X_i, \hat{\Theta}, \theta) \right| \leq \epsilon$ for some constant $\epsilon$ independent of $\theta, \ell, \ell'$ and $n$.

### 2.3. Estimating Kolmogorov with likelihoods

Fixing our notion of Kolmogorov memorization, we now describe how we can estimate $H^K$ in different setups. Note that exact calculation of Kolmogorov complexity is known to be uncomputable. However, we can still approximate it using the best available compression schemes. Below, we summarize how we approximate each term in our definition.

- $H^K(x \mid \hat{\theta})$: Here, $\hat{\theta}$ is the trained target model, which does not necessarily capture the true data distribution. Because compression rate is inherently tied to the likelihood under a predictive model (Shannon, 1950), we can easily estimate $H^K(x \mid \hat{\theta})$ using $p(x \mid \hat{\theta})$, the likelihood of $x$ under the target model.

- $H^K(x \mid \hat{\theta}, \theta)$: In this case, the compression algorithm has access to both target and reference models. We simply compute $\max\{p(x \mid \hat{\theta}), p(x \mid \theta)\}$. In practice, our choice of reference model is a larger model with the same architecture as $\theta$ trained for many steps on a much wider data distribution.

## 3. Model Capacity for Memorization

Unintended memorization provides us a principled way of measuring the precise number of bits a model $\theta$ knows about a datapoint $x$. If we add up the information for each

---
[2]The trained model and each data sample can be presented using $\ell$ and $\ell'$ bits respectively.

datapoint in a dataset, we can measure the total amount of bits a model knows about the dataset. And in cases where generalization is not possible because each datapoint is completely independent, we can estimate the **capacity** of a given model $\theta$ by summing per-datapoint unintended memorization.

### 3.1. Defining *model capacity*

We first formalize this notion of memorization capacity for a particular language model $\theta$. Capacity is the total amount of memorization that can be stored in $\theta$ across all its parameters.

*Definition* 5 (Capacity). Let $X$ be a distribution and $L: X \rightarrow \Theta$ a learning algorithm. We define the capacity of the learning algorithm $L$ to be

$$\mathrm{Capacity}(L) = \max_X \mathrm{mem}\big(X, L(X)\big)$$

We treat $X$ as an i.i.d. vector of $n$ coordinates whose marginals are the data distribution; the "size" of $X$ refers to $n$, the number of samples drawn.

When the model capacity is reached, $\mathrm{mem}(X, L(X))$ will no longer increase with dataset size. In practice, we can compute capacity by training to saturation on varying sizes of $X$ and computing the maximum memorization. By *saturation* we mean the regime where the training loss no longer improves with further iterations; we sidestep architecture- and hyperparameter-dependent saturation times by training all models for $10^6$ steps, well past saturation for every configuration we tested (10).

### 3.2. Measuring model capacity with synthetic sequences

In this section we measure the capacity of Transformer language models. Our goal is to instantiate multiple datasets and distributions and measure the memorization when training a single model $\theta$. Then, we take the maximum over all datasets to approximate the model's capacity. For instantiating our datasets, each token is uniformly sampled from a predefined set of tokens independent of the previous tokens.

To approximate $H^k(x \mid \theta, \hat{\theta})$, we can directly compute entropy under the trained model to calculate the shortest description of the dataset conditioning on $\hat{\theta}$. Subtracting the two, we can approximate the unintended memorization $\mathrm{mem}_U(X, L(X))$. Since the process for sampling the data is completely random, there is no generalization to be stored within $\hat{\theta}$. We formalize this with the following proposition:

*Proposition* 6 (generalization is zero for uniform random data). Let $\theta$ be the (fixed) data-generation model, let $X$ be sampled with respect to $\theta$, and let $\hat{\Theta}$ be a model trained on $X$. Then

$$\mathrm{mem}_I(X, \hat{\Theta}, \Theta) = 0,$$

and therefore $\text{mem}_U(X, \hat{\Theta}, \Theta) = \text{mem}(X, \hat{\Theta})$.

A proof is given in A.12.

Observe that when we sample synthetic sequences from a uniform distribution, we can compute their Shannon information exactly. Given a dataset size $N$, we construct a dataset of $N$ sequences, each of $S$ tokens. Given a vocabulary size $V$, we can calculate the total entropy of a dataset $x^i$ with such parameters by $H(x^i) = NS \log_2 V$. Then we calculate the compressed form $x^i$ using entropy under $\hat{\theta}_i$ to compute the code length and use this as an approximation of $H^K(x^i \mid \hat{\theta}_j)$. Then we calculate the $\text{mem}(x^i, \hat{\theta}_i) = H(x^i) - H^K(x^i \mid \hat{\theta}_j)$ and compute a model's capacity as the maximum amount of memorization over all datasets.

**Experimental details.** In accordance with Kaplan et al. (2020), we train models with the GPT-2 architecture (Radford et al., 2019) initialized from scratch. Our models have between 1 and 8 layers, hidden dimensions scaled from 32 to 512, and from 100K to 20M parameters. We train models for $10^6$ steps with a batch size of 2048 using the Adam optimizer. All models are trained on a single A100 GPU in bfloat16 precision, and we use gradient accumulation if a batch cannot fit in memory. Unless otherwise noted, we set vocabulary size $V = 2048$, sequence length $S = 64$ and vary only the number of points in a dataset. We train each model on each dataset size over five random seeds.

**Results.** We plot memorization across model and data sizes in Figure 1. This allows us to visualize unintended memorization amounts (y-axis) across dataset sizes (x-axis) grouped by model size (line color). We observe a striking plateau once a model reaches its capacity. Given the dataset is large enough, models exhibit an upper bound in net memorization, regardless of data size. Small datasets are completely memorized by all models with enough capacity.

We estimate the capacity of each model as the maximum amount of unintended memorization in bits measured across all dataset sizes. We then compare this capacity to the model size in Figure 11. Even at this small scale, we see a very smooth relationship between observed capacity and model parameters. We plot this relationship in Figure 11: under these settings, our models consistently memorize between 3.5 and 3.6 bits per parameter. This corroborates the findings of prior work such as (Roberts et al., 2020; Lu et al., 2024), which noticed that fact storage scales linearly with model capacity. Ours is a slightly larger estimate than Allen-Zhu & Li (2024), which estimated via quantization that models can store around 2 bits per parameter.

Since our models are learned via gradient descent, they are not guaranteed to find the global optima; thus, we are only measuring a lower bound on model capacity. We plot

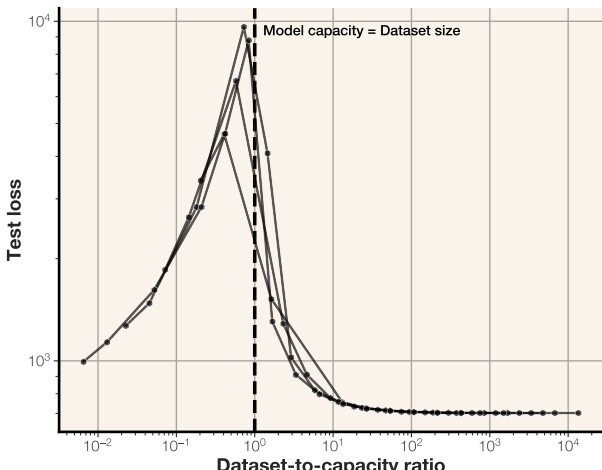

*Figure 3.* In our experiments on synthetic bitstrings, double descent occurs exactly when the dataset size begins to exceed the model's capacity, when unintended memorization is no longer beneficial for lowering the loss.

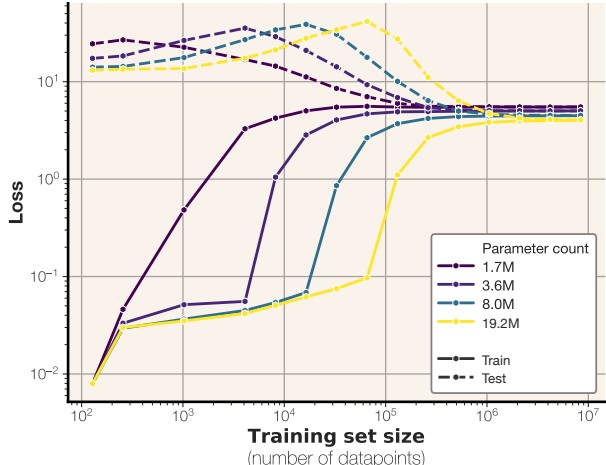

*Figure 4.* Train and test losses of different model and dataset sizes trained on text. Double descent occurs when dataset size exceeds model capacity.

model convergence throughout training in Figure 10. All datasets from 16,000 to 4M samples fall within a range of $3.56 - 3.65 \times 10^6$ bits memorized, indicating that our measurements are robust and we do not expect to memorize significantly more information by training for more steps. This finding confirms that capacity scales roughly with parameter count.

**How does precision affect capacity?** One natural question is how our estimates for $\alpha$ depend on the precision of language model training. Since all other experiments have been conducted in bfloat16 precision, we rerun our experiments in full fp32 precision to analyze the effect on capacity. Across model sizes, we observe a small increase in capacity, and an increase in $\alpha$ from 3.51 to 3.83 bits-per-parameter

| $n_{\text{layer}}$ | $d_{\text{model}}$ | Params | Capacity($\theta$) *[bits]* fp32 | bf16 | $\alpha$ *[bpp]* fp32 | bf16 |
|---|---|---|---|---|---|---|
| 1 | 32 | $8.04\times10^4$ | $3.39\times10^5$ | $3.16\times10^5$ | 4.23 | 3.93 |
| | 64 | $1.85\times10^5$ | $7.27\times10^5$ | $6.93\times10^5$ | 3.92 | 3.74 |
| | 128 | $4.69\times10^5$ | $1.71\times10^6$ | $1.69\times10^6$ | 3.65 | 3.61 |
| | 256 | $1.33\times10^6$ | $4.15\times10^6$ | $3.83\times10^6$ | 3.12 | 2.88 |
| 2 | 32 | $9.31\times10^4$ | $3.87\times10^5$ | $3.31\times10^5$ | 4.16 | 3.56 |
| | 64 | $2.35\times10^5$ | $9.60\times10^5$ | $9.27\times10^5$ | 4.08 | 3.94 |
| | 128 | $6.67\times10^5$ | $2.66\times10^6$ | $2.60\times10^6$ | 3.99 | 3.89 |
| | 256 | $2.12\times10^6$ | $8.49\times10^6$ | $7.76\times10^6$ | 4.01 | 3.66 |
| 4 | 32 | $1.18\times10^5$ | $4.65\times10^5$ | $3.99\times10^5$ | 3.92 | 3.37 |
| | 64 | $3.35\times10^5$ | $1.34\times10^6$ | $1.14\times10^6$ | 3.98 | 3.39 |
| | 128 | $1.06\times10^6$ | $4.02\times10^6$ | $3.75\times10^6$ | 3.78 | 3.53 |
| | 256 | $3.70\times10^6$ | $1.36\times10^7$ | $1.30\times10^7$ | 3.68 | 3.51 |
| 8 | 32 | $1.69\times10^5$ | $5.12\times10^5$ | $4.85\times10^5$ | 3.02 | 2.86 |
| | 64 | $5.35\times10^5$ | $2.05\times10^6$ | $1.71\times10^6$ | 3.83 | 3.19 |
| | 128 | $1.86\times10^6$ | $7.23\times10^6$ | $6.49\times10^6$ | 3.89 | 3.49 |
| | 256 | $6.86\times10^6$ | $2.71\times10^7$ | $2.51\times10^7$ | 3.96 | 3.65 |
| | | | | Mean ($\pm0.1$): | 3.83 | 3.51 |

*Table 1.* Model capacity estimates across different widths and depths in full and half-precision. Doubling precision from bfloat16 to float32 only increases model capacity from 3.51 to 3.83 bits-per-parameter.

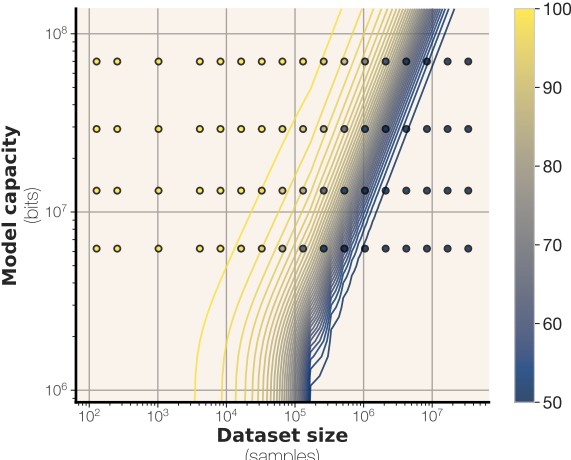

*Figure 5.* Scaling law curves for membership inference overlaid with empirical data, shown in circles.

on average. This is far less than the actual 2x increase in the bits of $\theta$, indicating that most of the extra model bits added when increasing precision from bfloat16 to float32 are not used for raw storage.

**Relationship to scaling laws.** A natural concern is whether our 3.6 bits-per-parameter measurement is a re-parameterization of established scaling laws (Kaplan et al., 2020). Kaplan-style laws predict held-out cross-entropy as a function of model size and training tokens, and the difference between two such predictions could in principle reproduce a memorization-like quantity. Our synthetic capacity measurement avoids this confound by construction: each capacity number is computed as the gap between the data's analytically known Shannon entropy ($NS\log_2 V$)

and the model's compression of its own training data. This is a property of training-set fit, not held-out generalization, and Kaplan's framework makes no claims about training cross-entropy in the memorization regime. Moreover, on uniform random data, held-out cross-entropy is $\log_2 V$ for any model that has learned the marginal — Kaplan's curve is flat by construction in this setting and so any reference-vs-target gap derived from it would be zero. The capacity we report comes from training-time compression beyond the entropy floor, a quantity orthogonal to Kaplan-style validation scaling.

## 4. Disentangling Unintended Memorization from Generalization

Our previous experiments analyzed the memorization and membership inference properties of synthetic bitstrings. We now turn to measuring memorization of text. Unlike randomly generated sequences, learning from text data is a mix of both unintended memorization (sample-level) and generalization (population-level).

**Experimental details.** We repeat the experiments from 3.2, substituting our synthetic datapoints for real text. We use the FineWeb dataset (Penedo et al., 2024) as it follows state-of-the-art deduplication practices. We use sequences of 64 tokens but perform an additional deduplication step to ensure perfect deduplication. We find careful deduplication extremely important for faithfully measuring extraction rates. As in the previous subsection, we pretrain models of varying sizes on different-sized text datasets and measure the unintended memorization of each model-dataset pair. In addition to memorization, we measure membership inference performance and compute exact extraction rates by greedily decoding prefixes of different lengths.

**Results.** We first observe that the sample-level unintended memorization increases with model parameters and decreases with training set size (Figure 4). When we measure unintended memorization with respect to an oracle reference model (Figure 2), memorization steadily increases as our smaller model learns more about the small training set than the oracle, then decreases as our model starts to generalize and perform worse than the (higher-capacity) oracle on average.

**On the role of the reference model.** The absolute scale of unintended memorization on real text depends on the reference model $\theta$, since a less capable reference yields higher per-token loss and therefore a wider gap to the target. This dependence does not undermine the framework: for any fixed reference, MEM$_U$ defines a consistent ordering across target models and dataset sizes, so relative comparisons ("does target $A$ memorize more than target $B$?") are

invariant to the choice of $\theta$ as long as it is held constant. We verify this empirically in A.13, where we re-score the same target runs against three reference models spanning a $6\times$ size range (124M, 355M, 774M, all FineWeb-matched): absolute MEM$_U$ values shift across references, but the qualitative shape of the curve is preserved. We use the largest reference (774M) throughout the main text for the tightest absolute estimates.

**Dataset-to-capacity ratio predicts double descent.** We observe from the train and test loss that for larger datasets the model only begins to generalize once its capacity is reached, which takes approximately $10^5$ samples, depending on parameter count. As in Nakkiran et al. (2019) we plot the ratio between the dataset size and model capacity (Figure 3). Unlike prior work, we can compute the exact dataset size and exact model capacity (based on our estimate of $\alpha$).

We clearly observe that evaluation performance decreases as the training set size nears model capacity, and then rapidly drops as the dataset capacity exceeds the capacity of the model. Our observations offer an intuitive explanation for double descent (Belkin et al., 2019; Nakkiran et al., 2019): double descent begins exactly when the data capacity exceeds the model capacity. Once the model can no longer memorize datapoints individually, it is forced to share information between datapoints to save capacity, which leads to generalization.

**Generalization explains nonzero extraction rates.** We measure extraction rates on the full training set and 10,000 non-overlapping test samples (Figure 16). We note that for 32-token prefixes, 100% are extractable for very small training set sizes; all extraction numbers decrease with training set size. When the dataset size grows sufficiently large, the extraction rate does not go to zero; however, it converges to nearly exactly the test extraction rate. In other words, when our deduplicated dataset grows sufficiently large, all successful training data extraction is attributable to generalization. This implies that extraction-based notions of memorization conflate intended and unintended memorization: a training sample being greedy-extractable is, in the large-data regime, no more evidence of unintended memorization than an unseen test sample being greedy-extractable.

### 4.1. Comparison of distributions memorized

**Distribution-level analysis.** Text sequences have very different properties than uniform synthetic bitstrings. We explore how two models of equal capacity spread their memorization across datapoints. We plot a histogram (Figure 8) of train and test compression rates of training data from both synthetic random bitstrings and text. Random training data follows a very normal distribution with a small amount of overlap between train and test compression rates. Text loss

is lower on average but more spread out, with low loss on some training points and a long tail of higher losses. There is much more overlap between the train and test loss distributions, which explains why membership inference is more difficult for text data.

**Which datapoints are most memorized?** Our distribution-level analysis indicates that unlike in the random-bitstring case, models trained on a large amount of text are able to memorize a small number of datapoints. Prior work has indicated that a large amount of this memorization can be due to duplicated training points (Lee et al., 2022) but our dataset is fully deduplicated so this cannot be an explanation in our case.

To quantitatively evaluate the number of rare words per document, we measure the TF-IDF of each training document, plotted vs. unintended memorization in Figure 9. We use the following equation for TF-IDF:

$$\text{TF-IDF}(d; \mathcal{D}) = \frac{1}{|d|} \sum_{w \in d} \log \frac{|D|}{tf(w, \mathcal{D})}$$

where $tf(d, \mathcal{D})$ indicates the total number of times word $w$ appears in dataset $\mathcal{D}$. Intuitively, a higher TF-IDF score for document $d$ indicates that $d$ contains more words that are rare in $\mathcal{D}$.

## 5. Memorization and Membership

Our training settings allow total control over the train and test data and come with perfect deduplication. This makes our setting ideal for studying the relationship between model size, dataset size, and membership inference success rate.

All of our membership inference results come from a standard loss-based membership inference (Yeom et al., 2018; Sablayrolles et al., 2019). The method is very simple: we set a cutoff loss value to predict whether a sample is or is not a member of the training dataset.

### 5.1. Membership in synthetic and text data

**Synthetic data.** For each of our models trained on synthetic data, we plot the success rate of the membership inference attack across dataset sizes. We show results in Figure 15. Above a certain dataset size, membership inference starts to fail in the average case. This finding indicates that if the dataset size is too large compared to the model, membership inference of an average training sample may not be possible.

**Text.** For each of our models trained on text, we use unused non-overlapping data from FineWeb to perform a standard loss-based membership inference (Yeom et al.,

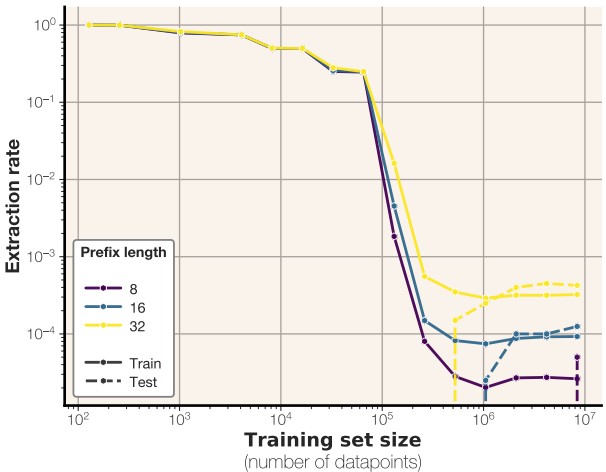

*Figure 6.* Extraction rates of 64-token training sequences across prefix lengths, for both train and evaluation.

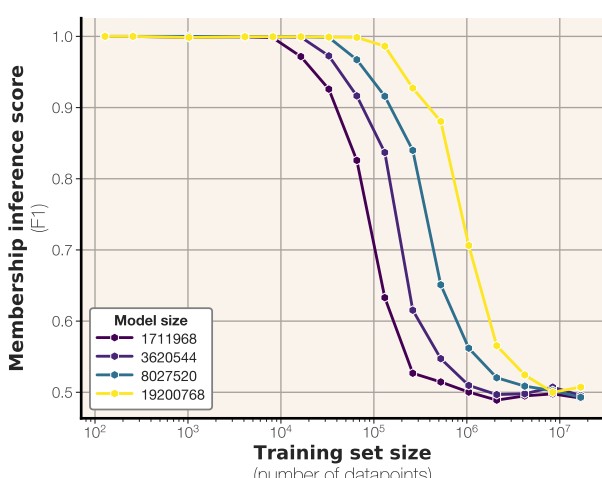

*Figure 7.* Membership inference F1 across dataset sizes. F1 score of 0.5 implies random guessing.

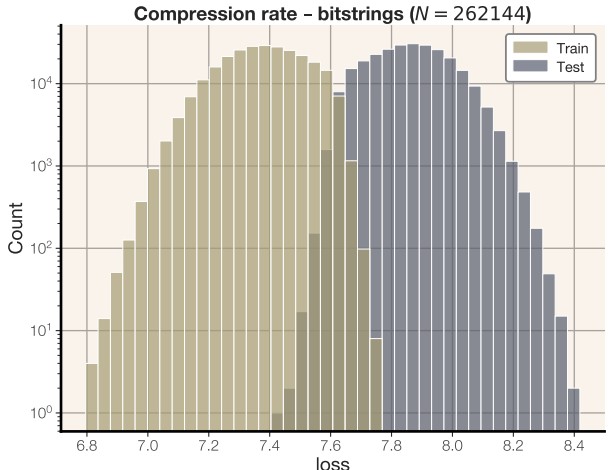

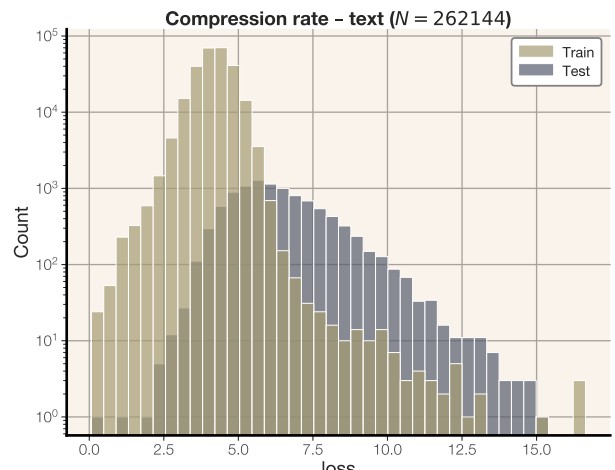

*Figure 8.* Distribution of compression rates for equal-sized transformers ($n_{\mathrm{layer}} = 4$, $d_{\mathrm{model}} = 128$) trained on $2^{14}$ sequences of equal-length random bitstrings (left) and text (right).

2018; Sablayrolles et al., 2019) and plot performance across dataset sizes (Figure 7). For a fixed model size, membership inference gets more difficult as the data size increases. When comparing membership inference to extraction (Figure 6), membership inference is strictly higher in every case; in some cases we can infer training dataset membership quite well (score of 0.97) with an extraction rate of 0.

### 5.2. Scaling laws for Membership

In this section we develop a set of predictive models for memorization. Specifically, we predict the F1 score of a loss-based membership attack given token count, number of examples, and model parameter count. We then validate our predictions on models from 500K to 1.5B parameters.

#### 5.2.1. FUNCTIONAL FORMS

We observe that for a fixed model capacity, membership inference follows a roughly sigmoidal form with respect to dataset size. The intuitive explanation is that membership inference is easy for large models overfit to tiny datasets, so its score begins at 1; as dataset size increases, differentiating train from test data by loss becomes more difficult, eventually decaying toward 0.5.

We reuse the data collected in our text experiments (Section 4) to solve for constants $c_1, c_2, c_3$ in the following equation:

$$\mathrm{Membership}_{F_1}(\theta, \mathcal{D}) = \frac{1}{2}\left(1 + c_1 \sigma\left(c_2 \frac{\mathrm{Capacity}(\theta)}{|\mathcal{D}|} + c_3\right)\right)$$

where $\sigma(x) = \frac{1}{1+e^{-x}}$.

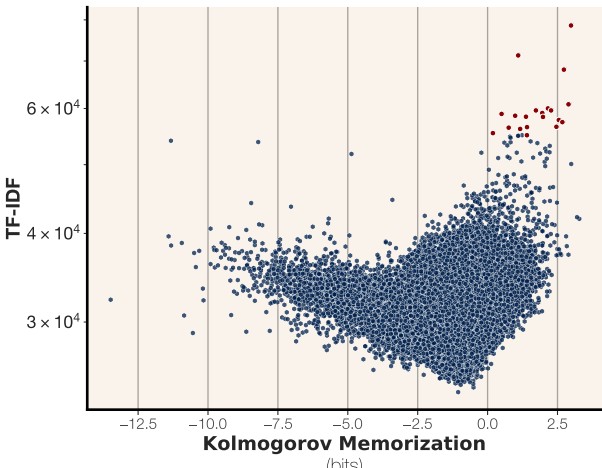

*Figure 9.* Unintended memorization vs. TF-IDF for all training points of a $20M$ param model trained past its capacity on $2^{16}$ sequences of English text. The training documents with rarest words are typically the most memorized.

**Limiting behavior.** We observe that as $|\mathcal{D}| \to \infty$, performance of our membership inference attack decreases to 0.5 (essentially random performance). For a model trained on an infinite dataset, our law predicts both membership inference and extraction to be impossible.

**Fitting.** We use a non-linear least squares solver to find optimal values for $c_1, c_2, c_3$. Solutions found are $c_1 = 1.34$, $c_2 = -0.034$, and $c_3 = -33.14$. We plot the scaling laws along with observed data in Figure 5. Although the sigmoidal function is slightly simplistic, our fit produces estimates within 1-2% of observations.

### 5.2.2. VALIDATION ON LARGER MODELS

We note that all contemporary language models trained with a tokens-per-parameter ratio of $10^2$ or higher would have a membership inference score of 0.5 according to our laws.

To validate our predictions, we train models with expected membership F1 scores of 0.55, 0.75, and 0.95. For model sizes we select GPT-2 small (125M params) and GPT-2 XL (1.5B params). Using our scaling law, we solve for the dataset size required to get the desired membership inference score for the given model size. We train models on the estimated dataset size and measure F1 score (Figure 5). Our predictions are generally within 1.5 points of the true F1 score. The accuracy of our results indicates that our empirical model of membership inference is relatively accurate and provides evidence for why membership inference attacks fail on models trained on extremely large datasets (Das et al., 2024; Duan et al., 2024; Maini et al., 2024).

## 6. Conclusion

We propose a new definition of memorization that allows us to measure the exact number of bits a model knows about a dataset. We use our definition to measure the capacity of modern transformer language models and analyze how measurements such as extraction and F1 score scale with model and dataset size. We also propose a scaling law for membership inference and validate it on larger models. Our results help further practitioner understanding of how language models memorize and what they might (or might not) be memorizing across model and dataset scales.

## Impact Statement

This paper investigates the capacity of language models to memorize their training data, proposing a formal framework to disentangle unintended memorization from generalization. Our goal is to provide a principled understanding of how much information models retain about individual training examples, and how this relates to model size, dataset size, and membership inference vulnerability. We believe this work contributes to the broader effort of understanding privacy risks in machine learning and can inform the development of training practices that better protect sensitive data.

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

## A. Appendix

### A.1. Related Work

**Language models and compression.** Shannon's source coding theorem (Shannon, 1948) first formalized the duality between prediction and compression. The connection between language modeling and compression was studied as far back as Shannon (1950), which observed that more accurate models of English can compress text in fewer bits. Other works note the connection between Kolmogorov complexity (Kolmogorov, 1965) and Shannon information in detail (Grunwald & Vitanyi, 2004). Delétang et al. (2024) investigate using modern transformer-based language models as compressors. We use compression as a tool to measure memorization in models.

**Language model capacity.** (Arpit et al., 2017) formalize the idea of *effective capacity* of a model and its training procedure; they also observe that both representation capacity and training time have a strong impact on empirical model capacity. Several other works measure language model capacity in the number of facts or random labels that can be memorized by a network such as an RNN (Collins et al., 2017; Boo et al., 2019) or transformer (Roberts et al., 2020; Heinzerling & Inui, 2021; Allen-Zhu & Li, 2024), sometimes under quantization. A few research efforts (Yun et al., 2019; Curth et al., 2023; Mahdavi et al., 2024; Kajitsuka & Sato, 2024) have developed theoretical estimates for the capacity of different model architectures, although none have yet scaled to multi-layer modern transformers. We are the first to measure a clear upper-bound in model capacity.

**Alternative definitions of memorization.** Unintended memorization is deeply related to the many other definitions of memorization proposed in the literature. We provide a detailed comparison in and Section A.3.

### A.2. Related Work: Definitions of Memorization

**Prior definitions of memorization.** Carlini et al. (2019) defined a string $m$ as memorized by a language model $\theta$ if the second half of $m$ can be generated greedily when prompting the model with the first half. Following this, Nasr et al. (2023) introduced *extractable memorization*, where model $\theta$ is said to memorize $m$ if an adversarial prompt $p$ can be found that generates $m$. Mireshghallah et al. (2022) and Schwarzschild et al. (2024) refined this definition by restricting $p$ to a certain number of tokens, preventing it from containing the entire $m$. However, even this definition has limitations: for example, generating the sequence "cat cat cat ... cat" with the prompt "repeat cat 1000 times" does not necessarily indicate memorization. Carlini et al. (2019) use perplexity or likelihood, one measure of the compressibility of a sequence, in an effort to distinguish

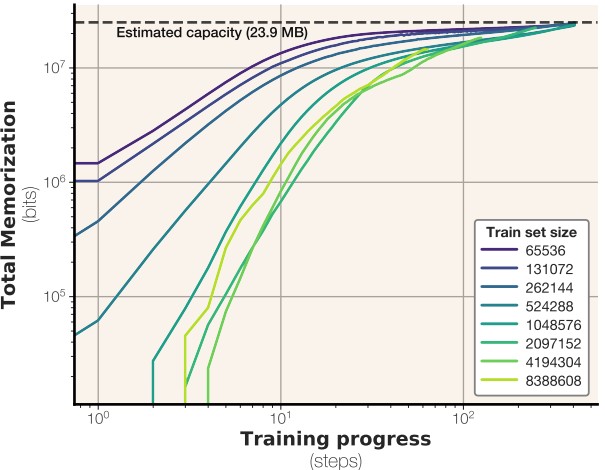

*Figure 10.* Bits memorized across training. This particular model is a GPT-style transformer with $6.86M$ parameters and a capacity of 23.9 MB.

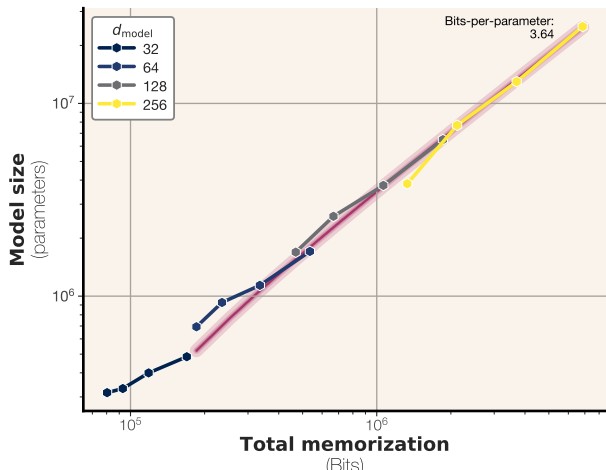

*Figure 11.* Capacity in bits-per-parameter for models trained on synthetic data. We estimate $\alpha = 3.64$ bits-per-parameter for GPT models trained in half precision.

highly memorized sequences from merely easy-to-compress ones. One additional definition of note is *counterfactual memorization* (Zhang et al., 2023), which measures the impact of a single datapoint on training; this can be seen as an instantiation of our definition where a different model of the same family is used as a reference model. Overall, all these works regarded memorization in terms that can be seen as forms of compression, although did not explicitly define it as such.

Finally, in an independent (but earlier) work of (Cohen et al., 2024), authors propose a theoretical definition for memorization also relying on Kolmogorov complexity. This notion is similar to us in using Kolmogorov complexity to measure information content but it does not separate unintended from intended memorization.

Some of our findings also relate to the discovery of *double descent* in machine learning (Belkin et al., 2019; Nakkiran et al., 2019) and language modeling (Xia et al., 2023), as well as general discussions of memorization and generalization in deep learning (Zhang et al., 2017; Tänzer et al., 2022).

Here, we discuss other definitions of memorization.

### A.3. Other notions of memorization

In this section we list multiple other notions of memorization and compare it with our definition. We specifically focus on why these notions do not satisfy all of our requirements.

- **Stability-based notions of memorization.** There are notions of privacy and memorization that deal with "stability" of the training algorithm to small changes in the training set. Most notably, differential privacy

(Dwork, 2006) considers the worst-cast drift of the model distribution when a single data point changes. Another notion of memorization in Feldman (2020) is based on the change of the model prediction on a point $x$, when we add the labeled pair $(x, y)$ to the training set of a classification/regression model. Both of these notions are crucially relying on the learning algorithm and how it behaves. Moreover, the definition of differential privacy is not ideal for our case because it is a worst-case definition and cannot be applied at sample/model level. While the notion of memorization in Feldman (2020) does not have this particular issue, it suffers from the fact that it only applies to classification models and mostly deals with the memorization of the association between the label ($y$) and input ($x$), and not the memorization of $x$ itself. These issues make these notions not ideal for our case.

- **Extraction-based memorization.** There are multiple works in the literature (Carlini et al., 2019; Mireshghallah et al., 2022; Nasr et al., 2023; Zhang et al., 2023; Carlini et al., 2023b; Schwarzschild et al., 2024) that define memorization of samples in language models based on how easy it is to extract that sample. Specifically, when trying to understand the extent of memorization of a sample $x$ in a model $\theta$ they measure some notion of complexity for the task of eliciting the model to output $x$. Although these notions are great in that they only take a model $\theta$ and a sample $x$, they still do not account for generalization. Considering our running example of the following training sample: "What is $2^{100}$? (A: $1, 267, 650, 600, 228, 229, 401, 496, 703, 205, 376$)", this will be identified as highly memorized by almost all of the extraction based notions of memorization.

Another issue with these definitions are that they are heavily dependent on the details of decoding algorithm. This is not ideal as we do not expect the memorization of a sample $x$ in a model $\theta$ to depend on the detailed parameters we use to generate samples using $\theta$.

The work of (Schwarzschild et al., 2024) in this category is the closest to ours. This work which is based on prompt-optimization, optimizes a short prompt $p$ to make the model elicit $x$, then it calls the sample $x$ memorized, if length of $p$ is less than $x$. Although this definition is close to our definition in using compression, it still does not account for generalization of the model. Moreover, it focuses on a specific way of compression through prompting. We posit that compression through prompting is an inferior compression scheme and can often lead to compression rates greater than 1.

- **Membership/attribute inference.** Membership inference (Shokri et al., 2017) and attribute inference attacks (Jayaraman & Evans, 2022) have been used for empirically measuring the privacy of machine learning algorithms. These notions which usually aim at approximating the stability notions of memorization are suffering from the same shortcomings. They rely heavily on the learning algorithm and the data distribution. Moreover, they fail at providing a sample level notion of memorization. For example, the obtained accuracy for membership inference attack is only meaningful in the population level. This is because various attack may have different true positives for membership, and the union of all these true positive across different attack may cover the entire training set, rendering it unusable as a sample level notion of memorization.

- **Data copying in generative models.** There are some interesting notions of memorization designed specifically for generative modeling where a generative model may output a certain portion of training samples (Bhattacharjee et al., 2023; Carlini et al., 2023a). These notions are similar to extraction based definition of memorization but they are more lenient in that they only require extraction of part of the training data. However, they still suffer from the same challenges as of extraction based definitions.

## A.4. Compression with language models beyond arithmetic coding

Shannon (1948) noted that the optimal compression method for a given source is one that assigns codes to symbols such that the average code length approaches the entropy of the source. Arithmetic coding (Pasco, 1977; Rissanen, 1976) is known to be one optimal way to compress text given a distribution over symbols; it was used in (Delétang et al., 2024) to compress text using modern language models.

Although arithmetic coding is known to be optimal for samples generated from the random process of choice, it may still be sub-optimal for cases where the compressed samples are correlated with the choice of random process. Specifically, in language modeling, the training data is highly correlated with the model itself and hence we might need to treat them differently. For instance, we know from previous work that the models behavior on training data points is different from random samples. A large portion of training data can be generated using greedy decoding (Carlini et al., 2023b; Liu et al., 2025) which is a behavior not expected for randomly sampled data. To this end, we design a new compression technique, a generalization of arithmetic coding.

**Ensemble compression.** Sampling from language models involve two key parameters $k$ for $top_k$ selection and $t$ for temperature. We design a compression method that sets these parameters adaptively. For instance, for cases where we know we can decode the next 100 tokens in a greedy fashion, we set $k = 1$ to reduce the bit length of arithmetic code. Changing the setup of the coding scheme itself requires a new token to be injected and wastes some number of bits, but it could still be beneficial for the code length. Our compression program uses dynamic programming to find the optimal code with injection of these new tokens in the middle of the text. Notably, our algorithm runs in time $O(n * T)$, where $n$ is the number of tokens and $T$ is the number of possible setups (combination of $t$ and $k$) that we allow.

**Relationship to probabilistic extraction.** Concurrent work by Hayes et al. (2025) studies memorization via *probabilistic* extraction: rather than requiring exact greedy reconstruction, they characterize the distribution of decoding outcomes (under varied temperature, top-$k$, top-$p$) that recover a training sample. Their framework is complementary to ours: any decoding strategy that compresses a training sample more tightly (whether deterministic temperature decoding, which we use, beam search, or the probabilistic-extraction methods of Hayes et al. (2025)) yields a tighter approximation to the Kolmogorov complexity $H^K(x \mid \hat{\theta})$ and therefore a tighter (more conservative) estimate of unintended memorization. Our compression-based definition is agnostic to the choice of decoder: a better compressor is, by construction, a strictly better estimator under our definitions, and improvements in extraction methodology directly translate to sharper memorization bounds.

| | $d_{emb}$ | $n_{layer}$ | $|\theta|$ | | $|D|$ | Predicted F1 | Observed F1 |
|---|---|---|---|---|---|---|---|
| GPT2-XL | 1600 | 48 | 1,556,075,200 | | 170,654,583 | 0.55 | $54.61 \pm 1.3$ |
| | | | | | 76,795,021 | 0.75 | $71.08 \pm 0.4$ |
| | | | | | 18,851,574 | 0.95 | $95.85 \pm 0.8$ |
| GPT2-Medium | 768 | 12 | 123,702,528 | | 13,566,442 | 0.55 | $53.44 \pm 1.1$ |
| | | | | | 6,104,935 | 0.75 | $65.69 \pm 0.6$ |
| | | | | | 1,498,634 | 0.95 | $97.98 \pm 0.3$ |

*Table 2.* Dataset sizes that our scaling law predicts will produce a given membership inference F1, along with empirical values.

| $S$ | Params. | Memorized | Expected | Error |
|---|---|---|---|---|
| 4 | $6.59 \times 10^5$ | $1.73 \times 10^5$ | $1.80 \times 10^5$ | 4.19 |
| 8 | $6.60 \times 10^5$ | $3.54 \times 10^5$ | $3.60 \times 10^5$ | 1.80 |
| 16 | $6.61 \times 10^5$ | $7.15 \times 10^5$ | $7.21 \times 10^5$ | 0.84 |
| 32 | $6.63 \times 10^5$ | $1.44 \times 10^6$ | $1.44 \times 10^6$ | 0.41 |
| 64 | $6.67 \times 10^5$ | $2.29 \times 10^6$ | $2.36 \times 10^6$ | 2.97 |
| 128 | $6.75 \times 10^5$ | $2.36 \times 10^6$ | $2.39 \times 10^6$ | 1.24 |
| 256 | $6.92 \times 10^5$ | $2.44 \times 10^6$ | $2.45 \times 10^6$ | 0.44 |

*Table 3.* Model capacity estimates across sequence length $S$, along with error (%).

| $V$ | Params. | Memorized | Expected | Error |
|---|---|---|---|---|
| 128 | $4.21 \times 10^5$ | $1.49 \times 10^6$ | $1.49 \times 10^6$ | 0.36 |
| 512 | $4.71 \times 10^5$ | $1.71 \times 10^6$ | $1.67 \times 10^6$ | 2.78 |
| 1024 | $5.36 \times 10^5$ | $1.95 \times 10^6$ | $1.90 \times 10^6$ | 2.70 |
| 2048 | $6.67 \times 10^5$ | $2.39 \times 10^6$ | $2.36 \times 10^6$ | 1.11 |
| 4096 | $9.29 \times 10^5$ | $3.13 \times 10^6$ | $3.15 \times 10^6$ | 0.47 |

*Table 4.* Model capacity estimates across vocab size $V$, along with error (%).

### A.5. How reliable are our linear estimates of capacity?

Instead of scaling the number of examples in a dataset, we scale model sequence length to adjust the size of a dataset. We use the following measurement for expected memorization of a model:

$$\text{mem}(X, L(X)) \approx \min(capacity(L), H(X))$$

we substitute our previous estimate of $\alpha = 3.642$ and ensure to adjust the parameter count for increases due to resizing the model's embedding matrices. We fix the number of training samples to 4096 and train a model with 2 layers and a hidden size of 128. Results are illustrated in Figure 12 and Table 3. Our predictions of total memorization are accurate, with an average error rate of 1.7% while scaling $S$ and 1.8% when scaling $V$.

### A.6. Additional memorization results

Our findings indicate that memorization of text data neatly plateaus near the model capacity just as in the synthetic data case. When the dataset size increases by a factor of $N$, the model divides its memorization between datapoints by an equal amount; the sum of memorization is measured to be constant, presumably at the upper bound of the model's capacity.

When the dataset is small enough for each model to fit – that is, below the capacity of the smallest model – we observe very similar performance between the models. For larger data sizes we notice an interesting trend: unintended memo-

rization increases with dataset size for to a point, presumably as a model fills its capacity with the available information, and then decreases, as the model replaces sample-level information with more useful, generalizable knowledge. A given model generalizes the most (and memorizes the least information about any individual sample) when the dataset is maximally large.

### A.7. Manual analysis of highly-memorized content

We clearly observe for samples with positive unintended memorization there is a strong correlation between trainset TF-IDF and memorization: examples with more rare words are more memorized. In particular, the sample with highest TF-IDF out of the whole training dataset (a sequence of Japanese words) has the third-highest measured memorization; even though this is just one out of $260,000$ training samples, the model can regurgitate the entire sequence given just a single token (囚). Out of the top twenty memorized sequences, all but three contain sequences of tokens from other languages (Japanese, Chinese, and Hebrew).

Manual analysis (Table 5) indicates that the most memorized datapoints have extremely rare tokens, typically ones not found in English.

### A.8. Scaling law fit

Here we demonstrate the fit of our sigmoidal scaling law to experimental data. We show points in tokens-per-parameter vs. fit in Figure 16. Although the sigmoidal function is slightly simplistic (the points do not perfectly fit the curve)

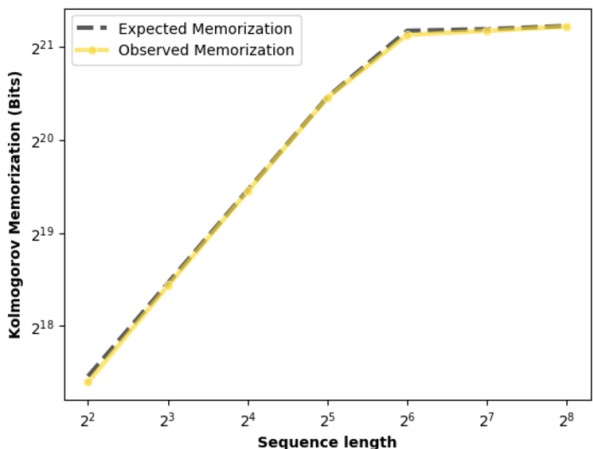

*Figure 12.* Model memorization across sequence lengths for a fixed-length dataset. Our predictions of total memorization are accurate, with an average error rate of 1.7%.

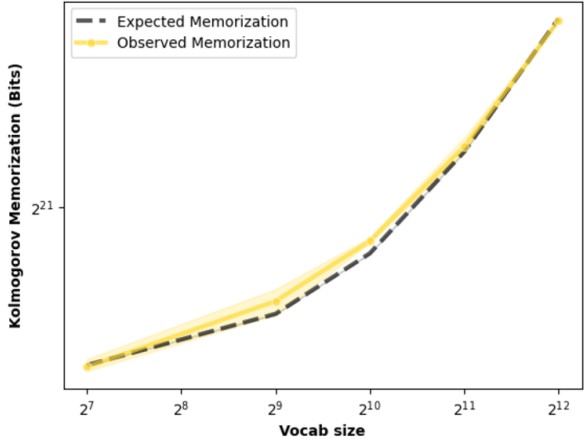

*Figure 13.* Model memorization across vocabulary size for a fixed-length dataset. Our predictions of total memorization are accurate, with an average error rate of 1.8%. Note that, we do not observe a capacity plateau, since increasing $V$ also increases parameters.

our fit produces estimates within $1 - 2\%$ of observations.

## A.9. Proofs

In the section we provide the proofs missing from the main body.

## A.10. Proof of Proposition 1

Here we prove Proposition 1.

We first state a lemma that gives the key super-additivity step.

*Lemma* 7. Let $X, Y, Z, T$ be four jointly distributed random variables, and suppose $X \mid T{=}t$ and $Y \mid T{=}t$ are independent for every $t$. Then

$$I\big((X, Y); Z \mid T\big) \geq I(X; Z \mid T) + I(Y; Z \mid T).$$

*Proof.* By the definition of conditional mutual information,

$$I((X, Y); Z \mid T) = H((X, Y) \mid T) - H((X, Y) \mid T, Z).$$

Since $X \mid T$ is independent of $Y \mid T$, the joint entropy decomposes: $H(X, Y \mid T) = H(X \mid T) + H(Y \mid T)$. By subadditivity of (conditional) entropy, $H((X, Y) \mid T, Z) \leq H(X \mid T, Z) + H(Y \mid T, Z)$. Combining,

$$I((X, Y); Z \mid T) \geq$$
$$H(X \mid T) + H(Y \mid T) - H(X \mid T, Z) - H(Y \mid T, Z)$$
$$= I(X; Z \mid T) + I(Y; Z \mid T). \qquad \square$$

*Proof of Proposition 1.* we have

$$\mathrm{mem}_U(X, \hat{\Theta}, \Theta) = I(X; \hat{\Theta} \mid \Theta)$$
$$= I((X_1, \ldots, X_n); \hat{\Theta} \mid \Theta).$$

Since the data is sampled i.i.d., all random variables in $\{R_i = [X_i \mid \Theta]\}_{i \in [n]}$ are independent.[3] Applying Lemma 7 inductively (grouping coordinates two at a time) gives

$$I((X_1, \ldots, X_n); \hat{\Theta} \mid \Theta) \geq \sum_{i \in [n]} I(X_i; \hat{\Theta} \mid \Theta)$$

which implies

$$\mathrm{mem}_U(X, \hat{\Theta}, \Theta) \geq \sum_{i \in [n]} \mathrm{mem}_U(X_i, \hat{\Theta}, \Theta).$$

On the other hand, we have

$$\mathrm{mem}_U(X, \hat{\Theta}, \Theta) = I(X; \hat{\Theta} \mid \Theta)$$
$$= H(\hat{\Theta}) - H(\hat{\Theta} \mid (X \mid \Theta))$$
$$\leq H(\hat{\Theta})$$

$$\square$$

## A.11. Proof of Proposition 4

*Proof.* We first state a Lemma about connection between algorithmic (kolmogorov) mutual information and mutual information.

*Lemma* 8. [Theorem 3.6 in Grunwald & Vitányi (2004)] Assume $(X, Y)$ be a pair of joint random variables. Let

---

[3]Note that $X_i$ themselves are not independent because they are sampled by first sampling an underlying model $\Theta$. However, they are conditionally independent once the underlying model $\Theta$ is given.

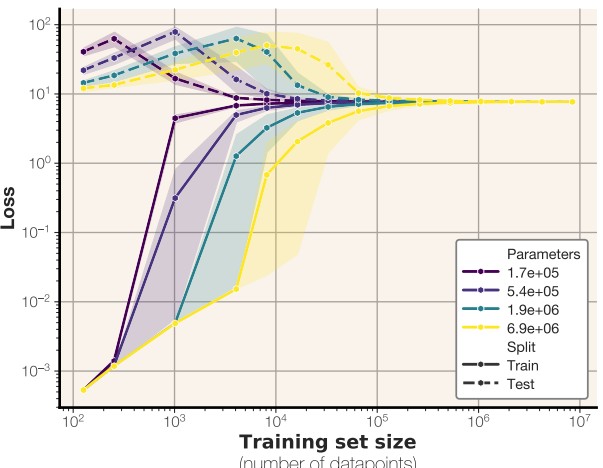

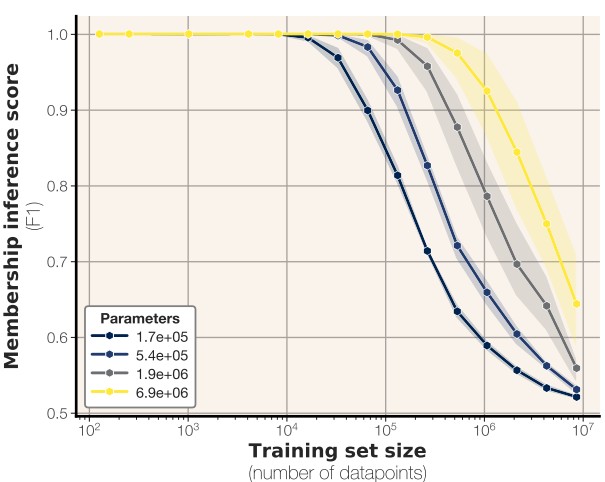

*Figure 14.* Train and test losses for different-sized language models trained on synthetic data.

*Figure 15.* Membership inference attack performance decreases with dataset scale. In the case of uniform synthetic data, membership inference performance never falls below $0.54$.

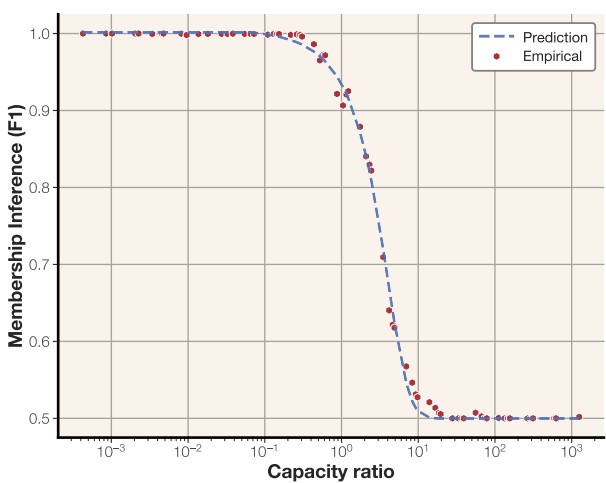

*Figure 16.* Our sigmoidal scaling law for membership inference fit to experimental data.

$f$ be the density function, $f(x, y) = \Pr[(X, Y) = (x, y)]$. Then we have

$$I(X; Y) - H_K(f) \leq \underset{(x,y) \sim (X,Y)}{\mathrm{E}}[I_K(x, y)]$$
$$\leq I(X; Y) + 2H_K(f).$$

Now we use this lemma to prove the statement of the Proposition. Let $f$ be a the density function for the joint distribution $(X_i \mid \theta, \hat{\Theta})$. That is $f(x_i, \hat{\theta}) = \Pr[X_i = x_i \text{ and } \hat{\Theta} = \hat{\theta} \mid \theta]$. Note that this function is independent of $n$ and $\theta$. By definition we have

$$\mathrm{mem}_U(X_i, \hat{\Theta}, \theta) = I(X_i; \hat{\Theta} \mid \theta).$$

Now using Lemma 8 we have

$$I(X_i, \hat{\Theta}; \theta) - H_K(f) \leq \underset{x_i \sim X_i \mid \theta}{\mathrm{E}}[I_K(x_i, \hat{\theta})]$$
$$\leq I(X_i; \hat{\Theta} \mid \theta) + 2H_K(f).$$

and this concludes the statement of Proposition by setting $\epsilon = 2H_K(f)$. Note that $f$ itself can be described using the description of the training algorithm and the sampling algorithm that uses $\theta$ to sample elements using $\theta$. This makes the shortest description of $f$ independent of independent of all $\ell, \ell', n$ and also $\theta$ itself. □

### A.12. Proof of Proposition 6

*Proof.* Let $\theta$ be the data-generation model. By definition,

$$\mathrm{mem}_I(X, \hat{\Theta}, \Theta) = H(X) - H(X \mid \hat{\Theta}) - H(X \mid \Theta) + H(X \mid \Theta, \hat{\Theta}).$$

| | Text | TFIDF | Memorization | Language |
|---|---|---|---|---|
| 0 | 人気エリアであるフォンニャに位置するRock & Roll Hostelは、ビジネス出張と観光のどちらにも最適なロケーションです。◆ | 78553.72 | 2.98 | Japanese |
| 1 | このトピックには0件の返信が含まれ、1人の参加者がいます。1年、6 ヶ月前に Dave Gant さんが最後の更新 | 71279.19 | 1.09 | Japanese |
| 2 | Label: Living Records\nDestroy All MonstersメンバーBen Millerによる自主レーベルからのソロCD-R。こちらは付属の抽象画をサウンド化したと | 68064.46 | 2.73 | Japanese |
| 3 | 《左傳》記「崔氏側莊公于北郭。丁亥，葬諸士孫之里，四翣，不◆ | 60820.46 | 2.89 | Chinese |
| 4 | 歡迎客人自備紋身圖案或要求本紋身店代客起圖，設計起圖須 | 60018.53 | 2.16 | Chinese |
| 5 | By 小森 栄治,向山 洋一\nRead Online or Download 中学の理科「総まとめ」を7日間で攻略する本 「◆ | 59625.40 | 2.27 | Japanese |
| 6 | 統合分析是將一些議題相關但彼此獨立的臨床實驗之研究結果(大◆ | 59624.37 | 1.73 | Chinese |
| 7 | 在SIA-Smaart Pro的Real-Time Module实时模块上，将功能扩展，实时显示相位和Fixed Point Per Octave（ | 59128.54 | 1.95 | Chinese |
| 8 | Progress in Intelligent Transportation Systems and IoT/M2M Communications: Markets, Standardization, Technologies\n出版日\|ページ情報\|英文 173 Pages\nインテリジェント交通システムお | 58953.67 | 0.50 | Japanese |
| 9 | English Title: Kingdom Hearts: Chain of Memories\nJapanese Title: キングダム ハーツ チェイン オブ メモリーズ — "Kingdom Hearts: Chain of Memories"\nAuthor: Tomoco Kanemaki\nIllustrator: Shiro | 58605.92 | 0.99 | Japanese |
| 10 | אשכול זה הועבר לארכיון. נא לשאול שאלה חדשה אם יש לך צורך | 58420.30 | 1.37 | Hebrew |
| 11 | 在《易經》里單数为阳, 双数为阴. 我曾怀疑马来西亚政府也会 | 58382.40 | 1.98 | Chinese |
| 12 | 「XXI c.—21世紀人」第3回企画展 三宅一生ディレクション\n21_21 DESIGN SIGHT 第３回企画展の | 57797.99 | 2.55 | Japanese |
| 13 | 无敌神马在线观看 重装机甲 睿峰影院 影院 LA幸福剧本\n时间：2020-12 | 57399.24 | 2.67 | Chinese |
| 14 | 季末小邪 回复 dgutkai: 楼主 您好 可以把项目源码发我吗? 可以付◆ | 56539.93 | 2.46 | Chinese |
| 15 | ◆בכל סדנא אפשר לזהות את הילדים שהוריהם מאפשרים חופש יצי | 56478.18 | 1.41 | Hebrew |
| 16 | Ακαδημαϊκές Δημοσιεύσεις Μελών ΔΕΠ σε άλλα Ιδρύματα >\n◆ | 56376.74 | 0.75 | Greek |
| 17 | Larry想和李华，还有她那些中国朋友多在一起玩儿，了解更多的中国文 | 56152.72 | 1.16 | Chinese |
| 18 | Mark 5:18 wrote:καὶ ἐμβαίνοντος αὐτοῦ εἰς τὸ πλοῖον παρεκάλει αὐ˙ | 55391.28 | 0.19 | Greek |
| 19 | בתחילה הייתי סקפטית לגבי השקעת כסף בשיווק אינטרנטי. | 55014.00 | 1.41 | Hebrew |

*Table 5.* Highest TF-IDF training examples from a $20M$ param model trained past its capacity on $2^{16}$ sequences of English text. All of the highest TF-IDF examples are considered memorized, and contain text from non-English languages (Japanese, Chinese, Hebrew, and Greek).

Since we first fix $\theta$ and then sample $X$ and $\hat{\Theta}$ with respect to $\theta$ (i.e., there is no stochasticity in $\theta$), we have $H(X) = H(X \mid \Theta)$ and $H(X \mid \hat{\Theta}, \Theta) = H(X \mid \hat{\Theta})$. Substituting,

$$\mathrm{mem}_I(X, \hat{\Theta}, \Theta) = H(X) - H(X \mid \hat{\Theta}) - H(X) + H(X \mid \hat{\Theta}) = 0.$$
□

### A.13. Reference-model size ablation

Our definition of unintended memorization on real text (4) depends on a reference model $\theta$ that approximates the true data distribution. The main paper uses a single oracle reference (Jones, 2024) (774M parameters, FineWeb-finetuned). In this section we report an ablation across three reference-model sizes from the same training-data distribution to characterize the sensitivity of $\mathrm{MEM}_U$ to this choice.[4]

**Reference models.** We score the same trained target models against three references, all matched to the FineWeb training distribution to isolate the size effect from data-distribution mismatch:

- gpt2 (124M), which we additionally finetune for one

---
[4]We thank the anonymous reviewers for suggesting this ablation.

epoch on FineWeb;

- gpt2-medium (355M), finetuned in the same way;
- gpt2-774M-fineweb-150B (774M), the baseline used in the main paper.

For each target run, we reuse the saved per-sample input ids and re-evaluate the loss under each reference model. unintended memorization is then $\mathrm{MEM}_U = \sum_{x \in X} \max(\mathcal{L}_\theta(x) - \mathcal{L}_{\hat{\theta}}(x), 0)$, identical to the main-paper definition.

**Data disjointness.** To keep the disjointness of the reference and target training data defensible, we partition the FineWeb sample-350BT split: target models train on FineWeb indices $[0, 5.1 \times 10^8)$ (102 prebuild shards of 5M deduplicated documents each, $\approx 441$M docs after global deduplication), and reference-model finetuning is restricted to the held-out tail $[5.1 \times 10^8, 5.18 \times 10^8)$.

**Sweep.** We train target models at $d_{\mathrm{model}} \in \{64, 128, 256\}$ with $n_{\mathrm{layer}} = 8$ across eight dataset sizes $|X| \in \{2^{14}, 2^{16}, 2^{18}, 2^{20}, 2^{21}, 2^{22}, 2^{23}, 2^{24}\}$, seed 1 for all cells (with seed 2 additionally at $d_{\mathrm{model}} = 128$ for an error-bar check). All other hyperparameters match the main

FineWeb sweep (seq_len $= 64$, lr $= 10^{-4}$, $10^5$ training steps, bfloat16).

**Results.** Figure 17 plots $\text{MEM}_U$ in bits as a function of dataset size, with one panel per target $d_{\text{model}}$ and one line per reference model. Across every cell, the measurement is monotone in reference size: smaller references yield larger $\text{MEM}_U$ values, because a less capable reference has higher per-token loss on FineWeb and therefore a wider gap between $\mathcal{L}_\theta$ and $\mathcal{L}_{\hat{\theta}}$. Equivalently, as the reference shrinks, the fraction of examples on which the reference outperforms the target shrinks, so fewer per-sample terms are clipped to zero by the $\max(\cdot, 0)$ operation. At the smallest dataset size ($|X| = 2^{14}$, $d_{\text{model}} = 64$), $\text{MEM}_U$ ranges from $\approx 1.0 \times 10^4$ bits (vs. 774M ref) to $\approx 2.6 \times 10^4$ bits (vs. 124M ref), a $2.5\times$ shift across a $6\times$ reference-size range. The absolute value of $\text{MEM}_U$ depends on the reference, as expected, while relative comparisons across target runs are preserved.

**Sensitivity panel.** Figure 18 re-plots the same data with reference size on the x-axis (one line per dataset size). For every target $(d_{\text{model}}, |X|)$ cell, $\text{MEM}_U$ decreases monotonically as the reference grows. The slope of the decrease is steeper for cells with more memorization (smaller $|X|$ or larger $d_{\text{model}}$), consistent with a multiplicative reference-size effect on the loss gap.

**Implications and a residual concern.** The ablation supports the main-paper claim that absolute $\text{MEM}_U$ values must be interpreted relative to a fixed reference, while relative comparisons across target runs are preserved. It is worth noting that when the reference shrinks toward the size of the target model, the assumption that the reference's loss bounds $\mathcal{L}_{\hat{\theta}}$ from above breaks for individual examples, and in that regime our framework can begin attributing generalization to $\text{MEM}_U$. Our smallest reference (124M) is still $\approx 6\times$ larger than our largest target ($\approx 20$M), but we do not extend the ablation into the ref $\approx$ target regime.

### A.14. Limitations

Our efforts to measure language model memorization come from a line of recent research to discover whether models have analyzed certain texts, and if so, how much. However, our main experimental contributions relate to the practice of training and evaluating language models, including a new perspective on the phenomenon of grokking (Nakkiran et al., 2019) and a new measurement of capacity. Our results are specific to the environment proposed and do not necessarily generalize to other datasets, architectures, or training setups.

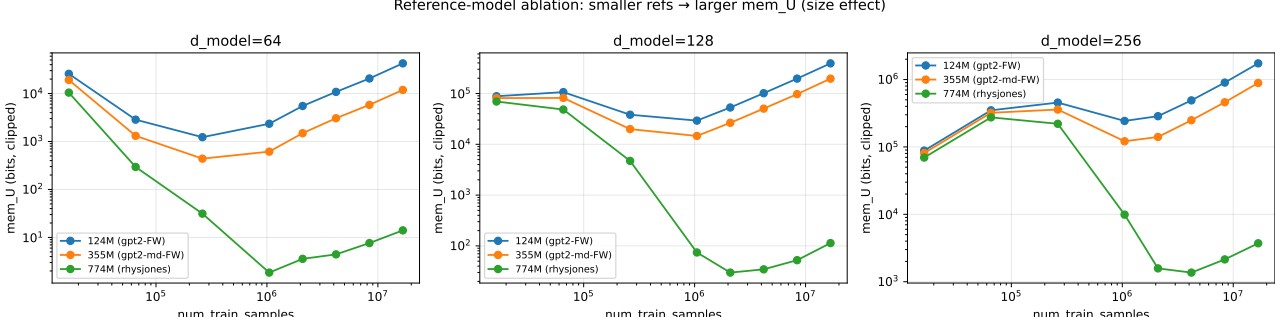

*Figure 17.* **Reference-model ablation.** MEM$_U$ (bits, clipped) vs. dataset size, one panel per target $d_{\text{model}}$. Lines are different reference-model sizes (all FineWeb-matched). Smaller references shift the absolute MEM$_U$ upward at every cell, while the qualitative shape of the curve is preserved.

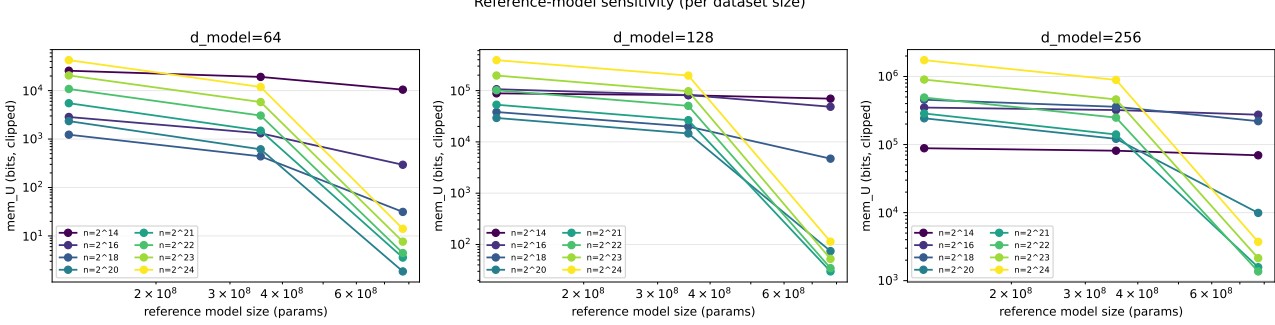

*Figure 18.* **Reference-model ablation, sensitivity.** For each target cell, MEM$_U$ as a function of reference-model size. Lines are different dataset sizes (color-coded). The monotone decrease in reference size confirms the size effect is consistent across all 24 target cells.

