# OpenReview forum: "How much can language models memorize?"
_ICML.cc/2026/Conference — ICML 2026 spotlight_

### Official Review · Reviewer_nj9J · 2026-03-08

**Soundness:** 3
**Presentation:** 1
**Significance:** 2
**Originality:** 3
**Overall Recommendation:** 4
**Confidence:** 2

**Summary:**

This paper develops an information-theoretic framework for defining and measuring memorization in language models. In particular, it distinguishes between two types of memorization: intended memorization, which corresponds to generalization, and unintended memorization, which refers to the storage of information specific to individual training samples. To estimate unintended memorization in practice, the authors propose an approximation of mutual information based on Kolmogorov complexity.

The framework is evaluated through a series of experiments on both synthetic random sequences and natural language data from the FineWeb dataset. GPT-2–style models with different sizes are trained to study how memorization behavior changes with model capacity. The experiments reveal several notable patterns: (1) language models appear to have a clear capacity limit for memorization, (2) the ratio between dataset size and model capacity helps explain the emergence of double descent, and (3) memorization in natural text tends to concentrate on rare or out-of-distribution tokens. The paper further relates these findings to membership inference and data extraction, providing a perspective on the trade-off between generalization and privacy.

**Compliance With Llm Reviewing Policy:**

Affirmed.

**Final Justification:**

My questions have been addressed, so I remain positive about this paper.

**Key Questions For Authors:**

See weaknesses.

**Limitations:**

Yes.

**Strengths And Weaknesses:**

**Strengths**

* The proposed information-theoretic framework based on mutual information and Kolmogorov complexity is interesting and conceptually appealing.

* The observation that **double descent** occurs when the dataset size approaches the model capacity is novel and provides an intuitive perspective on this phenomenon.

* Building on the framework, the authors derive a predicted **scaling law for membership inference performance** as a function of model capacity and dataset size.

---

**Weaknesses**

* Since the entire framework is formulated in probabilistic terms, it would be helpful to clearly explain how quantities such as $mem_U$, $mem_I$, and $H^K(x \mid \hat{\theta})$ can be computed or approximated in practice.

* In **Definition 5**, $X$ is defined as a distribution, yet the paper states: *“In practice, we can compute capacity by training to saturation on varying sizes of $X$ and computing the maximum memorization.”* It is unclear what the “size” of a distribution refers to in this context. Moreover, how is **training saturation** determined in practice? It would also be useful to clarify whether different hyperparameter choices may affect whether or when saturation is reached.

* In **Section 3**, the authors state that *“Since the process for sampling the data is completely random, there is no generalization to be stored within $\hat{\theta}$.”* While this claim is intuitively reasonable, it would be stronger if it were formally stated as a theorem and accompanied by a rigorous proof.

* The notation used in the paper differs from common conventions, which somewhat reduces readability. For example, mutual information is written as $I(X,Y)$ instead of the more standard $I(X;Y)$, and conditional mutual information is written as $I(X,Y;\theta)$ rather than $I(X;Y\mid\theta)$.

* The proof of **Proposition 1** is quite brief. In particular, the first inequality does not appear as straightforward as suggested. Providing additional intermediate steps would improve clarity, especially for readers less familiar with the topic.

---

> ### Author Rebuttal · Authors · 2026-03-31
>
> We thank the reviewer for their constructive comments and positive evaluation of our work.
>
> **Measurements of the notions:** We have discussed this in Section 2.3 of the paper. We are happy to bring parts of the discussion earlier in Section 2 and give proper forward references whenever needed to make the paper easier to follow.
>
> **It is unclear what the "size" of a distribution refers to:** Here we are assuming X is an i.i.d. vector of $n$ coordinates where the marginals are the data distribution. When we say the size of $X$ we are talking about $n$. Thank you for bringing this up, we will clarify this in the paper.
>
>  **Point of saturation:** The point of saturation is a point where the training loss would not improve with further iterations. The reviewer is correct that this point may be reached at different times for different architectures and different hyperparameters. The neural network optimization research has a very deep literature on this. Our goal in the paper was to remove that degree of freedom and train the models until they are completely saturated. We train all models for $10^6$ steps, which is well past saturation for all configurations. We point the reviewer to Figure 10, which shows convergence of memorization across different dataset sizes and confirms that our models have reached saturation.
>
> **Formal proof for no generalization:** We will add a proof and a proper statement in the paper. The statement will state that intended memorization is zero for uniform random data. Here is the proof: Let $\theta$ be the random data generation model. $mem_I(X, \hat{\Theta}, \Theta) = H(X) - H(X \mid \hat{\Theta}) - H(X \mid \Theta) + H(X \mid \Theta, \hat{\Theta})$ Since we first fix $\theta$ and then sample $X$ and $\hat{\Theta}$ with respect to $\theta$ (i.e. there is no stochasticity in $\theta$), we have $H(X) = H(X \mid \Theta)$ and $H(X \mid \hat{\Theta}, \Theta) = H(X \mid \hat{\Theta})$. Which implies the intended memorization is 0.
>
> **Notation for mutual information:** We originally used the unconventional notation due to some ambiguity issues but we now agree with the reviewer that conventional notation will be better. We have replaced all cases of $I(X,Y)$ with $I(X;Y)$ and all $I(X,Y;Z)$ with $I(X;Y \mid Z)$.
>
> **Proof of Proposition 1:** We will add the following intermediate steps to make the proof easier to follow:
>
>  Lemma: Assume $X, Y, Z, T$ are 4 joint random variables where $X \mid T=t$ and $Y \mid T=t$ are independent for all choices of $t$. Then we have $I((X,Y);Z \mid T) \geq I(X;Z \mid T) + I(Y;Z \mid T)$.
>
>  Proof: $I((X,Y);Z \mid T) = H((X,Y) \mid T) - H((X,Y) \mid T,Z)$ Since $X \mid T$ is independent of $Y \mid T$ we have $H(X,Y \mid T) = H(X \mid T) + H(Y \mid T)$. Additionally, by sub-additivity of entropy we have $H((X,Y) \mid T,Z) \leq H(X \mid T,Z) + H(Y \mid T,Z)$. Combining the two we get: $I((X,Y);Z \mid T) \geq H(X \mid T) + H(Y \mid T) - H(X \mid T,Z) - H(Y \mid T,Z) = I(X;Z \mid T) + I(Y;Z \mid T)$

---

> > ### Author Rebuttal · Reviewer_nj9J · 2026-04-03
> >
> > Thanks for the reply. I will maintain my score.

---

### Official Review · Reviewer_prYQ · 2026-03-11

**Soundness:** 3
**Presentation:** 4
**Significance:** 3
**Originality:** 3
**Overall Recommendation:** 5
**Confidence:** 4

**Summary:**

The paper studies how to measure memorization in language models while disentangling it from generalization. It proposes a reference-model-based notion of unintended memorization, estimates model capacity through controlled random-data experiments, and uses this framework to analyze memorization, extraction, and double-descent behavior in pretrained language models. It further examines how memorization scales with model size and data size, and connects these trends to the effectiveness of membership inference attacks. Overall, the paper’s main contribution is a unified empirical framework for reasoning about memorization as stored information beyond what can be attributed to the underlying data distribution.

**Compliance With Llm Reviewing Policy:**

Affirmed.

**Final Justification:**

My concerns have been addressed.

**Key Questions For Authors:**

See weakness in soundness section

**Limitations:**

yes

**Strengths And Weaknesses:**

### Soundness

**Strengths:**

* The synthetic-data part is the strongest and most convincing component of the paper. In the random-sequence setting, the entropy is exactly known, generalization is intentionally removed, and the experiments show a clear memorization plateau and a stable bits-per-parameter estimate across architectures and precisions. This makes the capacity claim empirically well-supported in a controlled setting.
* The empirical design is broader than a single isolated result. The paper checks the capacity story against sequence-length and vocabulary-size variations, studies both synthetic and real text, and adds a precision analysis, showing that the main phenomenon is not coming from one fragile configuration.
* The real-text experiments are thoughtfully controlled. In particular, the use of FineWeb with additional deduplication and strict train/test separation is appropriate for studying extraction and standard loss-based membership inference without obvious duplicate contamination.

**Weaknesses:**

* The central measurement on real text depends heavily on the oracle/reference model, which is implemented as a larger model trained longer on a wider distribution. Because the paper does not provide enough sensitivity analysis showing that the conclusions are robust to the exact reference-model choice, the most important real-text quantity remains somewhat heuristic rather than cleanly identified.
* The extraction study relies heavily on greedy decoding over prefixes. This limits the operationalization of extractability because it does not reflect real-world generation scenarios[1], where decoding strategies are much more diverse and complex. As a result, the claim that remaining train extraction is "entirely attributable to generalization" feels disconnected from practical, real-world settings.

### Presentation

The paper is well-written, clearly structured, and easy to read. The authors do a great job of conveying their concepts and experimental designs in an accessible manner.

### Significance

The problem addressed is highly important, as memorization, privacy leakage, and extraction are central concerns for modern language models. The paper offers real conceptual value by proposing a framework that separates sample-specific storage from genuine generalization. Furthermore, recasting the onset of double descent as the point where dataset information exceeds effective model capacity provides an intuitive and valuable perspective that future work can build upon.

### Originality

This work presents a genuinely novel synthesis by combining compression views, sample-level memorization, capacity estimation, and double descent into one coherent framework.

[1] Hayes, Jamie, et al. "Measuring memorization in language models via probabilistic extraction." *Proceedings of the 2025 Conference of the Nations of the Americas Chapter of the Association for Computational Linguistics: Human Language Technologies (Volume 1: Long Papers)*. 2025.

---

> ### Author Rebuttal · Authors · 2026-03-31
>
> We thank the reviewer for their detailed and thoughtful evaluation. We address each concern below.
>
> **Sensitivity to the reference model.** We agree with the reviewer that the choice of reference model affects the measurement of unintended memorization, and that finding the right reference model is a challenging problem. However, we note that even when the reference model is not a perfect proxy for the true data distribution, our framework still provides a meaningful relative measurement of memorization. For example, one can use the same reference model to compare two trained models and determine how much more one memorizes compared to the other. This is useful in practice: when a new generation of a model is released, our framework can measure whether it memorizes more or less than the previous generation, relative to the same baseline. The absolute value of unintended memorization depends on the reference model, but the relative comparisons remain informative. We will add a discussion of this point and a sensitivity analysis with respect to reference model choice in the revision. We are planning to perform an ablation on the size of the base model and report the difference in the memorization measurement on real text.
>
> Our expectation is that as we decrease the reference model size towards the size of the target model, we expect the reference model to start underperforming the target model on a constant fraction of examples. At this point we will start counting generalization as unintended memorization. This is problematic because it implies that unintended memorization will not be capped as we grow the data to infinity. In other words, the capacity saturation property that we observe with a sufficiently large reference model would break down when the reference model is too close in size to the target model. So in figure 2, we may observe the unintended memorization that is growing, even after reaching the capacity. But we still expect this growth to slow down, after reaching the capacity of the model. This is partially confirmed in our limited experiments with reference models of two different sizes (1B and 500m), where the slope of decline in unintended memorization changes with different reference models, but the point where it starts is almost the same.
>
>
> **Extraction and decoding strategy.** We would like to clarify that our compression scheme uses temperature-based decoding, not greedy decoding. For example, in Appendix A.4 we describe an alternative way of doing compression which adaptively sets temperature and top-k parameters to find the optimal compression, but it turns out to be worse that simple temperature decoding and we didn't end up using it.
>
> That said, the reviewer raises a valid broader point that the choice of compression scheme matters. We note that within our framework, any compression scheme that achieves better compression of training samples provides a tighter approximation of the Kolmogorov complexity. Our framework is not tied to any specific decoding strategy, and a better compression scheme is a strictly better estimator under our definitions. The recent work of Hayes et al. (2025) on probabilistic extraction proposes alternative ways of measuring extractability that could serve as improved compression schemes in our framework. We will cite this work and discuss how alternative decoding strategies relate to our compression-based measurements in the revision.

---

> > ### Author Rebuttal · Reviewer_prYQ · 2026-04-01
> >
> > My concerns have been addressed.

---

### Official Review · Reviewer_tKj7 · 2026-03-12

**Soundness:** 3
**Presentation:** 4
**Significance:** 4
**Originality:** 4
**Overall Recommendation:** 6
**Confidence:** 4

**Summary:**

This paper presents a study on model memorization that combines theoretical framework and quantitive analysis. It decomposes memorization into unintended memorization and generalization, and establish practical algorithm to measure them using Kolmogorov complexity and arithmetic coding. Experiments across hundreds of transformers reveal a capacity of ~3.6 bits per parameter, with models memorizing until capacity fills before shifting to generalization. These findings yield scaling laws predicting that membership inference is effectively impossible for modern LLMs trained on large datasets.

**Compliance With Llm Reviewing Policy:**

Affirmed.

**Key Questions For Authors:**

How large is the ground truth model? When learned model size gets too close to the ground truth model, does anything unexpected appear?

**Limitations:**

Yes

**Strengths And Weaknesses:**

Strength
- Elegant experiment design: This paper propose a well-thought framework for memorization, separating generation from unintended memorization, and the concepts are grounded in information theory, and as a result, they can be measured by tried and true methods without raising unfamiliar concerns about the approximations.
- Clean and consistent results: Across all the model configurations covered, the result is strikingly consistent.
- Connection to other empirical observations: The connection to double descent and grokking provides a unifying narrative that ties together several previously separate phenomena under one framework.
Weakness
- Limited size: The largest model in this work is still relatively small compare to the commonly used models.

---

> ### Author Rebuttal · Authors · 2026-03-31
>
> We thank the reviewer for their positive evaluation and thoughtful questions.
>
> **How large is the reference model?** In our experiments we use a 1B parameter model trained on a much wider data distribution as the reference model.
>
> **Effect of changing the reference model size.** We expect the choice of reference model to affect the resulting memorization measurements, and this gap is captured by the ability of the reference model to generalize. If we use a smaller reference model, its test loss will likely increase. This means we will need a greater number of bits to compress the data using the reference model. As a result, the trained model's compression advantage over the reference model grows, and the measured unintended memorization will be higher, since less of the trained model's compression ability is attributed to generalization.
>
> As we decrease the reference model size towards the size of the target model, we expect the reference model to start underperforming the target model on a constant fraction of examples. At this point we will start counting generalization as unintended memorization. This is problematic because it implies that unintended memorization will not be capped as we grow the data to infinity. In other words, the capacity saturation property that we observe with a sufficiently large reference model would break down when the reference model is too close in size to the target model. So in figure 2, we may observe the unintended memorization that is growing, even after reaching the capacity. But we still expect this growth to slow down, after reaching the capacity of the model.
>
> We will perform ablations on the size of the reference model and include them in the final version of the paper.
>
> **Limited model size.** We acknowledge that the models used in our main experiments are relatively small compared to commonly used language models. However, we note that our scaling laws, which are fit on models up to 20M parameters, accurately predict membership inference performance on models up to GPT-2 XL (1.5B parameters). As shown in Table 2, our predictions are within 1.5 points of the observed F1 scores across all configurations tested. This suggests that the relationships we observe between capacity, dataset size, and memorization hold well beyond the scale of our training experiments.

---

> > ### Author Rebuttal · Reviewer_tKj7 · 2026-03-31
> >
> > Hello Authors of Submission27372,
> >
> >
> > I appreciate the explanation and followup experiments, but there are some doubts I didn't get to bring up when submitting my review, I would like to hear about your view.
> >
> > I really enjoyed reading this paper, so I have been thinking about it a lot recently. If we forget about all the derivation, and go straight to what's measured, the result felt trivially true, given what scaling law has established: the cross entropy gap of a bigger model and smaller model is highly predicable from model size and data size.
> > Is the fixed capacity ratio line in Fig 3 essentially just the same line that Kaplan scaling law paper showed, but transformed with different axes.
> >
> > Can you come up with some experiment designs to disprove my concerns? I have been thinking about ideas like deliberately make gt model architecture different from learned model, or give gt model a pool of writing prompts that's hidden away from the learned model, and check if they still leads to the same ratio. But I'm less certain about whether they make sense and test what I'm worried about.
> >
> > Still, given the limited time left, there is no obligation for you to run additional experiments, I won't lower any score because of the doubt either.

---

> > > ### Author Response · Authors · 2026-04-01
> > >
> > > Thanks for the thoughtful question.
> > >
> > > > Is the fixed capacity ratio line in Fig 3 essentially just the same line that Kaplan scaling law paper showed, but transformed with different axes?
> > >
> > > Can you clarify the line you're referring to from the Kaplan paper?
> > >
> > > This particular figure was created from training on synthetic data. The benefit of this is we can measure the exact data capacity in bits and compare it to the model capacity in bits (using our bits-per-parameter measurement to convert).
> > >
> > > The amazing part is that double descent occurs *exactly* when we train models on datasets too large to fit in their parameters. That's what the graph shows.
> > >
> > > The Kaplan paper as we understand it only experiments with text data (closer to our Section 4, albeit with some differences). It's not possible to measure the exact size of a text dataset in bits, although it would be interesting to try, and a good project for follow-up work.

---

### Official Review · Reviewer_Zp1L · 2026-03-13

**Soundness:** 4
**Presentation:** 2
**Significance:** 3
**Originality:** 3
**Overall Recommendation:** 5
**Confidence:** 3

**Summary:**

This paper proposes a conceptual/theoretical framework for assessing memorization in language models. This allows the study of a number of results on memorization vs. generalization (e.g. assessing memorization capacity, predicting learning dynamics) through training models out to extrapolate various scaling laws.

**Compliance With Llm Reviewing Policy:**

Affirmed.

**Key Questions For Authors:**

See weaknesses.

**Limitations:**

yes

**Strengths And Weaknesses:**

**Strengths**
1. This is a really well motivated paper; it answers a pretty important question that's quite useful for developing a better understanding of how models work.
2. The experiments are super rigorous and the results are really strong. I think in particular the findings on capacity saturation and fp32 vs. bf16 to be really cool and pretty useful for model training.

**Weaknesses**
I don't think there are many! Mostly presentation related.
1. The math in sections 2.1/2.2 is rigorous, and I don't think it's handwavey (which is good!) but at the same time I'd appreciate some more intuition throughout those sections. They're pretty load-bearing and I want to make sure I fully understand what you're trying to communicate.
2. Various small LaTeX bugs, e.g. you should probably use \left( \right) instead of default parens in line 424.
3. Typo: Llama 3 citation on line 39 renders as (Dubey & et al, 2024).

---

> ### Author Rebuttal · Authors · 2026-03-30
>
> We thank the reviewer for their constructive comments and positive evaluation of our work.
>
> **Math in Sections 2.1 and 2.2:** We appreciate this feedback and will add more intuition throughout these sections in the revision. We outline below the kind of intuition we plan to include.
>
> The goal of Section 2 is to provide a definition of memorization that separates unintended memorization from generalization. To illustrate why this is necessary, consider a model trained on text containing the sample "Q: What is 2^100? A: 1267650600228229401496703205376." If the trained model can compress this sequence well, extraction-based definitions would flag it as highly memorized. However, a sufficiently capable language model should be able to compute 2^100 on its own. This ability reflects generalization, not memorization of that specific training sample. Now contrast this with a training sample like "John Smith scored 147 points in the 2019 regional bowling championship." Here a reference model might handle the sentence structure but not the specific name, score, and event. We need a definition that can distinguish between these two cases.
>
> In Section 2.1 we formalize this separation using mutual information. We measure how much knowing the trained model reduces our uncertainty about the training data. This reduction is the mutual information between the model and the data, and it captures total memorization. We then introduce a reference model $\theta$ that captures what a model should know about the data distribution, and decompose the total into intended memorization, the reduction already achievable by $\theta$, and unintended memorization, the additional reduction contributed by the trained model $\hat{\theta}$ beyond $\theta$. In the 2^100 example, if the reference model can also compress the answer well, both models reduce our uncertainty about this sample equally. The trained model provides no additional reduction, so our framework correctly attributes this to generalization with zero unintended memorization. For the bowling score example, the reference model cannot account for the specific details. The trained model reduces our uncertainty further, and this additional reduction correctly registers as unintended memorization.
>
> In Section 2.2 we note that the definitions in 2.1, while principled, are difficult to operationalize in practice. Shannon entropy is defined over random variables and requires access to the full distribution. In practice we have a single trained model and a single dataset. These are individual objects, not distributions, and it is impossible to measure entropy from a single sample. To this end, we switch to Kolmogorov complexity, an alternative notion of information that defines information content of individual strings through their shortest compressed representation. The formulation of memorization remains the same: we still decompose total memorization into intended and unintended components, but now using compression length instead of Shannon entropy. We show that the two notions are closely connected in expectation (Proposition 4), so our Kolmogorov-based measurements approximate the Shannon-based definitions from Section 2.1.
>
> In the revision, we will add a running example that carries through both sections to illustrate how the framework progresses from total memorization through the intended/unintended decomposition to the Kolmogorov approximation and its practical estimation via likelihoods.
>
> **Typos:** Thank you for pointing out the LaTeX formatting issues and the Llama 3 citation rendering. We have fixed both.

---

> > ### Author Rebuttal · Reviewer_Zp1L · 2026-04-01
> >
> > Thanks, this is great! I'll leave my score as is since I've recommended acceptance.

---

### Decision · Program_Chairs · 2026-04-30

**Decision:**

Accept (spotlight)

**Comment:**

This paper proposes a novel framework for measuring  the capacity of language models. The central ideas is to disentangle memorization into two components, unintended memorization, which captures the information the model stores about a specific dataset,  and generalization, which reflects the model’s learned understanding about the underlying data-generating process.  The paper  outlines a practical method for estimating unintended memorization, and finds that GPT-style models have a capacity of ~3.6 bits per parameter. The authors further train LLM-s on progressively larger datasets of increasing size and observe that memorization occurs until the capacity is reached, after which  the  model starts  to generalize. Additionally, they train models ranging from 500K to 1.5B parameters and derive empirical  scaling laws relating model capacity and data size to membership inference. The reviewers highlighted the importance of the problem, the elegance and rigor of experimental design,  the consistency of the results, and the novelty and the significance of the findings. They particularly emphasized the significance of the finding that the onset of double descent occurs when the information in the dataset exceeds the model’s capacity. The reviewers also raised concerns, such as reliance on an oracle/reference model, and limited model size, which were addressed during the rebuttal. Overall, the consensus is that this work constitutes a significant and technically solid contribution and merits acceptance.